# Hot Air-Assisted Radio Frequency (HARF) Drying on Wild Bitter Gourd Extract

**DOI:** 10.3390/foods11081173

**Published:** 2022-04-18

**Authors:** Chang-Yi Huang, Yu-Huang Cheng, Su-Der Chen

**Affiliations:** 1Department of Biotechnology and Animal Science, National Ilan University, Number 1, Section 1, Shen-Lung Road, Yilan City 26041, Taiwan; huaallen@gmail.com; 2Department of Food Science, National Ilan University, Number 1, Section 1, Shen-Lung Road, Yilan City 26041, Taiwan; rin520520ever@gmail.com

**Keywords:** wild bitter gourd, ultrasonic extraction, radio frequency (RF), drying, diabetic mice

## Abstract

Wild bitter gourd (*Momordica charantia* L. var. *abbreviata* S.) is a kind of Chinese herbal medicine and is also a vegetable and fruit that people eat daily. Wild bitter gourd has many bioactive components, such as saponin, polysaccharide, and protein, and the extract is used to adjust blood sugar in patients with diabetes. The objective of this study was to investigate simultaneous hot air-assisted radio frequency (HARF) drying and pasteurization for bitter gourd extract, and then to evaluate its effects on blood sugar of type II diabetic mice. The results showed that the solid–liquid ratio of the wild bitter gourd powder to water was 1:10 and it was extracted using focused ultrasonic extraction (FUE) for only 10 min with 70 °C water. Then, 1 kg of concentrated bitter gourd extract was mixed with soybean fiber powder at a ratio of 2:1.1. It was dried by HARF, and the temperature of the sample could reach above 80 °C in only 12 min to simultaneously reduce moisture content (wet basis) from 58% to 15% and achieve a pasteurization effect to significantly reduce the total bacterial and mold counts. Type II diabetic mice induced by nicotinamide and streptozocin (STZ) for two weeks and then were fed four-week feeds containing 5% RF-dried wild gourd extract did not raise fasting blood glucose. Therefore, the dried powder of wild bitter gourd extracts by HARF drying had a hypoglycemic effect.

## 1. Introduction

Wild bitter gourd (*Momordica charantia* L. var. abbreviata S.), also called wild bitter melon, is the Taiwanese endemic species of bitter gourds and features effective hypoglycemic components, such as saponin, polysaccharide, and peptide. Diabetic patients can consume fresh bitter gourd juice to reduce their blood glucose concentration and prevent postprandial hyperglycemia [1]. Bitter gourd saponins also promote glycogen storage and insulin secretion [2] and are involved in activating AMP-activated protein kinase phosphorylation to regulate energy metabolism [3]. Bitter gourd proteins can bond with insulin receptors to regulate blood sugar metabolism [4], and inhibit the activities of α-glucosidase and α-amylase to impede the degradation of starch and lower glucose content [5].

Traditional extraction methods used for plant extraction include mechanical stirring [6], boiling, and Soxhlet extraction [7]. Although traditional extraction methods are relatively inexpensive, they are subjected to the following limitations: lower extraction efficiency, longer extraction time, and larger solvent consumption [8]. In order to overcome these problems and to improve the extraction efficiency, ultrasonic extraction (UE) can be used. UE has three combined advantageous effects: cavitation, pressure, and thermodynamics. When ultrasonic waves propagate in the extraction liquid, ultrasonic vibration instantly causes the pressurization and decompression of the liquid, pushing the medium and generating cavitation. When countless small vacuum bubbles in the extraction liquid burst, high temperature and strong impact are instantly generated; therefore, the components of the extract can be separated to achieve an effective extraction efficiency. UE is a simple and inexpensive technology that can be easily applied to reduce solvent consumption, the extraction temperature, and time required to improve the extraction efficiency. Focused ultrasonic extraction (FUE) can promote and enhance biologically active ingredients from the plant in the extraction solvent. Therefore, the FUE method yields the highest total phenol content, total flavonoid content, total antioxidant capacity, and DPPH free radical scavenging capacity [9].

The plant extracts are mixed with encapsulates, such as maltodextrin, gum, or soy protein; they are directly sprayed under pressure to raise the surface area of the droplets using high-temperature hot air drying (over 100 °C), which is commonly used in the pharmaceutical industry. Most of the heat of hot air is used to enhance moisture evaporation, and the temperature of the dried powder can be controlled at 60~70 °C. The bitter gourd aqueous extract is mixed with encapsulate (maltodextrin and gum) as an encapsulating agent in 1:1.5, 1:2, and 1:3 ratios, and studied by spray drying [10,11,12]. The optimal inlet and outlet temperatures of spray drying have been determined to be 140 °C and 80 °C, respectively [11].

Moreover, most plant extracts are sterilized at a high temperature to avoid excessively high bacterial counts or mold counts prior to the final freeze-drying procedure. Traditional freeze drying uses a hot plate to conduct heat into the frozen food for ice sublimation. Freeze-dried encapsulated extracts feature a lower degradation and longer shelf life compared with spray-dried samples [13]. Although the quality of the freeze-dried product is good, the process is time consuming and energy consuming, which results in the manufacturing process being expensive.

Radio frequency (RF) waves are part of the electromagnetic spectrum. Three RF frequencies (13.56, 27.12, and 40.68 MHz) are allowed to be used in science, medicine, and industry by the US Federal Communications Commission (FCC) to avoid interference with other communication systems. The mechanisms of RF heating are due to the migration of ions and the rotation of polarized molecules, resulting in friction that generates heat. RF heating is an effective thermal treatment for various food and agricultural products used for different purposes, such as disinfestation, pasteurization, thawing, enzyme inactivation, and drying [14,15].

When RF waves passe through food, the water molecules in the food produce dipole friction to generate heat, causing the food to heat up quickly. RF energy leads to simultaneous water evaporation, pasteurization, and hot air-assisted water vapor transfer. Therefore, hot air-assisted radio frequency (HARF) drying can overcome the drawbacks of traditional hot air drying, which is both time and energy consuming [15]. Moreover, the developed RF vacuum drying method has been successfully applied for the drying of kiwi fruit for lower-temperature RF heating [16].

RF treatment has been increasingly studied in food drying [14] and pasteurization [17]. However, there remains a lack of studies on the use of HARF heating to dry the extracted solution directly. This may be due to the high dielectric loss factor of the extract solution, which induces an exceeding RF power and leads to an unstable process. The main barrier of HARF drying is the non-uniform heating, especially causing areas with large cold spots and overheating in parts of the high moisture materials, which results in undesirable qualities in the final products [18]. Therefore, a two-stage drying process was developed to overcome the difficulties of RF drying. For example, one study showed the moisture of mango slices was reduced from 88% to 40% by the first stage of hot air drying at 60 °C for 5 h, then the HARF drying was applied in the second stage of drying to further reduce the moisture content from 40% to 18% within 45 min. Therefore, the length of time needed for combination hot air and HARF drying is apparently lower than that of hot air drying (8 h) or vacuum drying (7 h). The overall quality of mango slices after the two-stage drying process was better than that of mango dried by hot air drying and close to that of mango dried by vacuum drying [17]. Furthermore, HARF drying has also been used in the final stage of drying in pre-dried carrot cubes (250 g) to reduce the moisture content from 40% to 10% using 4 h of drying [19].

Various encapsulating agents, such as polysaccharides, gum, lipids, and proteins, are added into the aqueous extract during spray and freeze drying [13]. Currently there is no research regarding drying aqueous extract using HARF drying. Therefore, soybean fiber powder was added into the extract solution as an encapsulating agent for reducing moisture content to solve the HARF drying problem. The objective of this study was to use HARF drying for the simultaneous drying and pasteurizing of the concentrated bitter gourd extract, and then to evaluate the hypoglycemic effect by adding 5% in the feed for type II diabetic mice.

## 2. Materials and Methods

### 2.1. Materials

Hot air dried wild bitter gourd was obtained from Asakusa Agriculture Processing Co. (Hualien, Taiwan). Bovine serum albumin (BSA), Coomassie brilliant blue G-250, 1,1-Diphenyl-2-picryl hydrazyl (DPPH), ascorbic acid (Vitamin C), glacial acetic acid synthetic and antioxidant butylated hydroxyl anisole (BHA) were bought from Sigma Chemical Co. (St. Louis, MO, USA). Ginsenoside Rg1 was bought from ChromaDex, Corp. (Los Angeles, CA, USA). Aerobic count plate (3M Petrifilm 6400), yeast, and mold count plates (3M Petrifilm 6477, 500 EA/CS), and papain (2000 FCCU/mg, Decken Biotech, Taichung, Taiwan) were acquired. Soybean fiber powder was bought from Prime Creative International CO., LTD (Hsinchu, Taiwan). Experimental male mice (BALB/c strain), MFG feed, and BETA chip were purchased from BioLASCO Taiwan Co. Ltd (Taipei, Taiwan). Streptozocin (STZ) and Nicotinamide were purchased from Sigma-Aldrich^®^ (St. Louis, MO, USA)

### 2.2. Equipment

A focused ultrasonic extractor (20k Hz, 1400 W, Ever Great Ultrasonic Co., New Taipei City, Taiwan), spectrophotometer (Model U-2001, Hitachi Co., Tokyo, Japan), benchtop centrifuge (HERMLE Z300, Gosheim, Germany), mini-protein system (Bio-Rad, CA, USA), RF with hot air equipment (40.68 MHz, 10 kW, Yh-Da Biotech Co., LTD., Yilan, Taiwan), oven (Channel DCM-45, Yilan, Taiwan), digital pocket refractometer (Pocket, 3810, PAL-1, ATAGO Corp., Tokyo, Japan), and multifunctional infrared thermometer (Testo104-IR, Hot Instruments Co., LTD., New Taipei, Taiwan) were employed.

### 2.3. Focused Ultrasonic Extraction (FAE) and Hot Water Extraction (HWE) of Wild Bitter Gourd

The wild bitter gourd was ground using a 60 mesh, and 50 g of the ground wild bitter gourd was mixed with 500 mL of reverse osmosis (RO) water. Samples were extracted from the solutions at 20 kHz, 1400 W, and 70 °C for 10 min by FAE or using 100 °C hot water extraction for 1 h.

The solid content of the extract was converted from the measurement of °Brix by an analogue refractometer. The extract was centrifuged at 6000 rpm for 5 min and stored at 4 °C for analysis.

### 2.4. Hot Air-Assisted Radio Frequency (HARF) Drying Extract from Wild Bitter Gourd

The extract obtained by FUE from wild bitter gourd was concentrated six times under vacuum conditions to raise the solid content from 4°Brix to 24°Brix before HARF drying for time-saving, energy-saving, and high-efficiency purposes.

In this study, soybean fiber powder was used as encapsulate and mixed with extracts from wild bitter gourd to overcome the issue of the sample being too wet. The samples were prepared with 1 kg of wild gourd concentrated extract were homogeneously mixed with soybean fiber powder in three ratios of (A) 2:1, (B) 2:1.1, and (C) 2:1.2, respectively. Then, they were put into polypropylene (PP) plastic containers (31.3 × 6.9 × 20.5 cm^3^) and treated at a HARF equipment (Figure 1). HARF drying was carried out at 10 cm gap between the parallel electrode plates and 100 °C hot air. The surface temperatures of the samples were measured at three points (at the center and 5 cm on either side of the center), and the weights by balance were measured at the time interval of 30 s to obtain the temperature profile and drying curve.

### 2.5. Analytical Methods

#### 2.5.1. Protein Content Measurement

Testing was conducted using the Bradford protein-binding assay by mixing 800 µL of wild bitter gourd extract with 200 µL of protein assay reagent. The solution was left idle for 10 min of reaction time. The absorbance at the 595 nm wavelength was measured three times, and the mean value was calculated. The BSA standard curve was employed to determine the protein concentration in the sample.

#### 2.5.2. Total Saponins Measurement

The saponins of the sample was measured by vanillin-perchloric acid colorimetry, and this experiment was modified from the method of [20]. The 50 mL extract was mixed well with 70% ethanol for 10 min of reaction. Then, it was centrifuged (6000 rpm, 5 min) to obtain 0.2 mL of the supernatant, into which 0.20 mL of 5% vanillin-glacial acetic acid solution and 0.80 mL of perchloric acid were then added, and the mixture was heated in a 60 °C water bath for 15 min. When it was cooled, 5 mL of glacial acetic acid was added for 20 min reaction and measured the absorbance at a wavelength of 548 nm three times. The mean value was calculated. The ginsenoside (Rg1) standard curve was employed to determine the total saponins concentration in the sample.

#### 2.5.3. DPPH Free Radical Scavenging Ability Test 

For this test, 2 mL of the supernatant was evenly mixed with 2 mL of 0.2 mM DPPH MeOH solution and left idle away from light at room temperature for 30 min. The absorbance at the 517 nm wavelength was measured and applied to the following equation to calculate the DPPH scavenging ability. [21] The result was compared with that for the control group comprising 5 mg/mL of ascorbic acid and BHA.

#### 2.5.4. Microbiological Test

The extracts (1 mL) were each diluted 10, 100, and 1000 times with sterilized water. Subsequently, 0.5 mL of each diluted extract was cultured at 35 °C for 72 h and counted on an aerobic count plate for a total bacteria test, yeast count, and mold count. The results were observed and the bacterial colonies were counted.

### 2.6. Hypoglycemic Effect of Wild Bitter Gourd Extract on STZ-Induced Mice

#### 2.6.1. Experimental Animal and Raising Environment

Twenty-four 6-week-old BALB/c male mice (BioLASCO Taiwan Co., Ltd., Yilan, Taiwan) were quarantined for 1 week and acclimated in polyethylene cages at the experimental animal room of National Ilan University, where the temperature was 22 ± 2 °C, the relative humidity was 60 ± 5%, and the light cycle was 12 h light/12 h dark. They were fed with free water and MFG feed. The wood shavings (Beta Chips) and shredded aspen shavings were purchased from BioLASCO Taiwan Co. Ltd. and changed every 3 d. The body weights of all mice were measured before the experiment, and weekly thereafter. The fasting blood glucose of all mice was measured by ACCU-CHEK active blood glucose meter after one week of adaptation to being randomly assigned to control and treatment groups (8 mice in each group) based on body weight.

#### 2.6.2. Preparation of Feed 5% Bitter Gourd Extract Powder

The concentrated concentrate of bitter gourd extract was mixed with soybean fiber powder in the ratio of 2:1.1, and the bitter gourd extract was dried by RF energy, added to the original mouse feed and crushed, and a little sterile water was added to make a thick shape and then mixed homogeneously; and after cold air drying, the mouse feed containing 5% of the bitter gourd extract powder was made.

#### 2.6.3. Regulated Blood Glucose Function Evaluation Method

According to the evaluation of blood sugar regulation method by the Ministry of Health and Welfare, Executive Yuan. Except for the normal group, all mice were intraperitoneally injected with Nicotinamide (230 mg/kg) and then STZ (75 mg/kg) for two weeks. Then, the experimental mice were divided into three groups of eight mice each (normal, STZ, and STZ-fed extract). The STZ mice were fed a diet containing 5% of dried extract powder from wild bitter gourd bitter, and the normal diet mice and STZ mice were used as control groups. Because each group of eight mice was not fed one mouse in one cage, it was difficult to obtain the accurate data on water and feed intake of each mouse, and the data are not shown here. During the test period, the changes in body weight of each mouse were recorded, and fasting blood glucose values were measured at the initial week and final fourth week to provide a basis for evaluation of the hypoglycemic effect.

### 2.7. Statistical Analysis

The test results were expressed as mean ± SD and analyzed using the Statistical Package for Social Science 14.0 (SPSS Inc., Data Statistical Analysis Corporation, Chicago, IL, USA). Differences between the data were examined through one-way analysis of variance, and the significance of the differences was investigated through Duncan’s multiple range test (α = 0.05).

## 3. Results and Discussion

### 3.1. Focused Ultrasonic Extraction (FUE) of Wild Bitter Gourd

Table 1 shows the extraction qualities of the wild bitter gourd powder mixed with water at a solid to liquid ratio of 1:10. The extraction ratios of the extract from ultrasonic extraction at 70 °C for 10 min and conventional hot water extraction at 100 °C for 60 min were 22.4% and 21%, respectively. The contents of total saponins and total proteins extracted by hot water were higher than those of the extract by ultrasonic extraction. However, the scavenge DPPH free radicals extract by ultrasonic was 79.2% and was significantly higher than 59.6% of hot water extract.

Garude et al. [9] contended that the phenol and flavonoid contents in bitter gourd peel extracts by ultrasonic extraction were higher than those in extracts by hot water extraction. The ultrasonic water extract of fresh bitter gourd exhibited a strong antioxidant ability of scavenging DPPH free radicals and a higher inhibition of α-amylase and α-glycosidase activity [22].

Therefore, traditional high-temperature long-duration extraction degraded the antioxidant properties of the bioactive components; however, short-duration and low-temperature ultrasonic extraction could avoid damaging the bioactive components and antioxidant properties of the extract. Compared with traditional extraction, FUE took only 10 min; therefore, it could also achieve the time-saving and energy-saving effects. Considering the cost of extraction, previous studies have recommended that the optimal conditions for the focused ultrasonic extraction of wild bitter gourd powder are mixed with water in a solid–liquid ratio of 1:10 at 70 °C for 10 min to obtain 6°Brix. The extraction yield has been shown to be higher than that of bitter gourd aqueous mixture made using a shaking water bath with a solid–liquid ratio of 1:20 at 40 °C water for 15 min, where only a 2% total solid content was obtained [10].

### 3.2. Hot Air-Assisted Radio Frequency (HARF) Drying of Extract from Wild Bitter Gourds

Therefore, the moisture content decreased with the increasing level of additional soybean fiber powder. Figure 2 shows the temperature profiles and drying curves of these three samples during 10 kW radio frequency (RF electrode gap of 10 cm) combined with 100 °C hot air drying. With the same loading of 1 kg of sample, the drying time was reduced from 14 to 10 min as the ratio of mixed soybean fiber powder increased. In addition, the surface temperature of the sample quickly reached above 70 °C during the first 4 min of RF heating, then the temperature of sample gradually increased to 80 °C, as most RF-induced heat was absorbed for latent heat of water evaporation.

Table 2 shows that the period of constant drying rate appeared during the RF drying. Increasing the soybean fiber powder addition from (A) 2:1 to (B) 2:1.1 and (C) 2:1.2 caused higher drying rates of 36.881, 43.836, and 53.968 g/min, respectively. The moisture contents of the final products were 14.2%, 16.7%, and 12%, respectively. The HARF drying time may be extended by 1 min more to further reduce the moisture content for storage. The level of additional soybean fiber powder in the original extract solution was about 33~37.5% in HARF drying. The encapsulate (maltodextrin and gum powder 1:1) addition in the bitter gourd aqueous extract was about 20~25% [11,12] by spray drying with inlet and outlet temperatures of 140 °C and 80 °C, respectively [10]. In this study, the final temperatures of samples were also near 80 °C, and the drying time was much less than that of spray drying and freeze drying, because the feed rate of spray drying was only 10 mL/min [10] and freeze drying required at least 24 h.

### 3.3. HARF Pasteurizing of Wild Bitter Gourd Extract

The analyses of total bacterial counts are shown in Table 3. The original total bacterial counts for ultrasonic extract from the wild bitter gourd and soybean fiber powder were 523,000 CFU/mL and 9400 CFU/g, respectively. However, after HARF drying, the counts became only 30 CFU/mL in the 2:1.1 ratio of mixture of wild bitter gourd extract to soybean fiber. Both the mold and yeast counts were significantly reduced after the HARF drying. It was found that RF induced heating for the samples to achieve a pasteurization effect (above 70 °C) within the first 4 min. Then, the rest time of HARF heating could keep the temperature in a range of 70~80 °C due to the induced heat absorbed as latent heat for water evaporation. The moisture contents of the samples were also reduced from 58% to 15%, and the mold and yeast counts were significantly reduced. Therefore, HARF heating could proceed simultaneously for the drying and pasteurizing of the extracts.

### 3.4. Hypoglycemic Effect of Radio Frequency Dried Wild Bitter Gourd Extract

The average water intake of STZ mice was 2.8 g at week 0, which was lower than that of normal mice (3.8 g); there was no significant difference between the three groups after one week of feeding. Furthermore, the average feed intake of STZ mice was 3 g at week 0, which was slightly lower than the 3.2 g of normal mice. Similarly, there was no significant difference in average feed intake of about 3.2 g between the three groups after one week of feeding. The STZ mice had litter less body weight, especially in the STZ-fed extract group (Table 4) and the higher blood glucose 105~107 mg/dL than normal mice (74.88 mg/dL) (Table 5). The mice were injected with nicotinamide and STZ to induce them to become type II diabetic mice, which had higher blood glucose levels at beginning. However, it did not completely destroy the pancreas; the consequences would thus be minimal.

After four weeks of treatment, the average body weight of these three groups of mice increased to 24.78~25.35 g with no significant difference (*p* > 0.05). However, after one week of feeding with RF-dried bitter gourd extract, the average body weight of the STZ-fed extract group mice was the same as that of the normal group. The average body weight of the STZ-fed extract group mice for the next three weeks was even slightly higher than that of the normal group (*p* > 0.05) (Table 4).

After four weeks of feeding, the blood glucose of mice in the normal group was 90 mg/dL as the feeding time increased, while the blood glucose of mice in the STZ group increased from 107 mg/dL to 115 mg/dL, but the blood glucose of mice in the STZ-fed extract group did not increase and remained at 105 mg/dL. Although the 5% RF-dried extracts from the STZ mice did not reach the blood glucose level of normal mice, they could control the increase in blood glucose. This suggests that the dried powder of ultrasonic extracts by HARF had a hypoglycemic effect.

The MCP was isolated from the water-soluble polysaccharide form of the bitter gourd and was orally administered once a day though water after 3 d of alloxan-induction type II diabetic mice at 300 mg/kg body weight for 28 d, exhibiting a hypoglycemic effect. Hence, MCP can be incorporated as a supplement in health care food, drugs, and/or combined with other hypoglycemic medicines [23].

## 4. Conclusions

In this study, wild bitter gourd powder at a solid–liquid ratio of 1:10 was extracted with ultrasonication for 10 min with hot water at 70 °C. The total protein and saponin contents were 1.198 mg/g and 0.288 mg/g, respectively. The ability of these samples to scavenge DPPH free radicals was better than that of the 100 °C hot water extracts. HARF for drying the mixture of 1 kg of concentrated wild bitter gourd extract and soybean fiber powder (at a ratio of 2:1.1) took only 4 min to achieve the pasteurization effect due to the temperature being above 70 °C, and the moisture content of the sample was reduced from 58% to 15% by a total of 12 min of HARF heating. This indicates that using ultrasonic extraction and HARF drying to prepare extracts for the herb and pharmaceutical industries can save time and energy. Type II diabetic mice were fed with a free diet containing 5% RF-dried wild bitter gourd extract for four weeks and their blood sugar levels did not rise. As a result, the dried powder of ultrasonic wild bitter gourd extracts by HARF heating also had a hypoglycemic effect.

## Figures and Tables

**Figure 1 foods-11-01173-f001:**
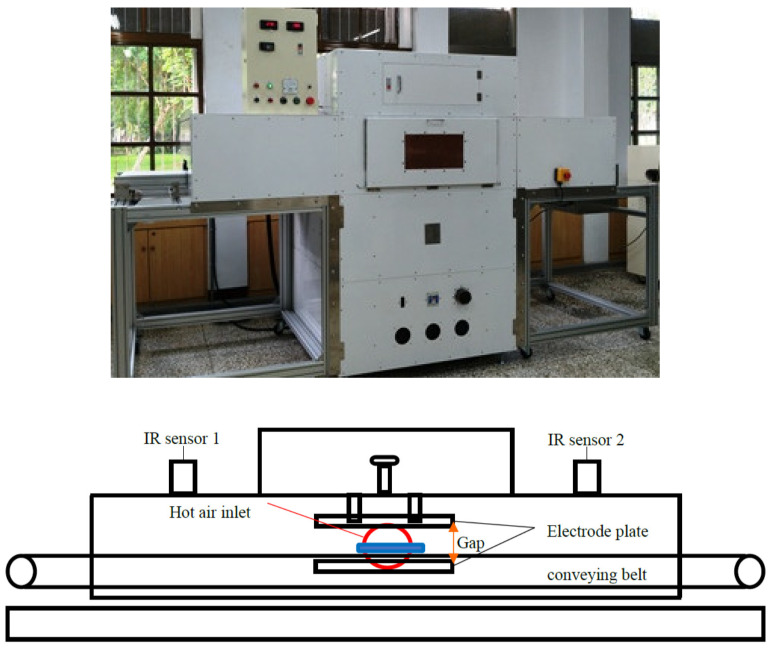
Schematic view of 10 kW, 40.68 MHz continuous radio frequency (RF) (10 kW, 40.68 MHz) with hot air heating system.

**Figure 2 foods-11-01173-f002:**
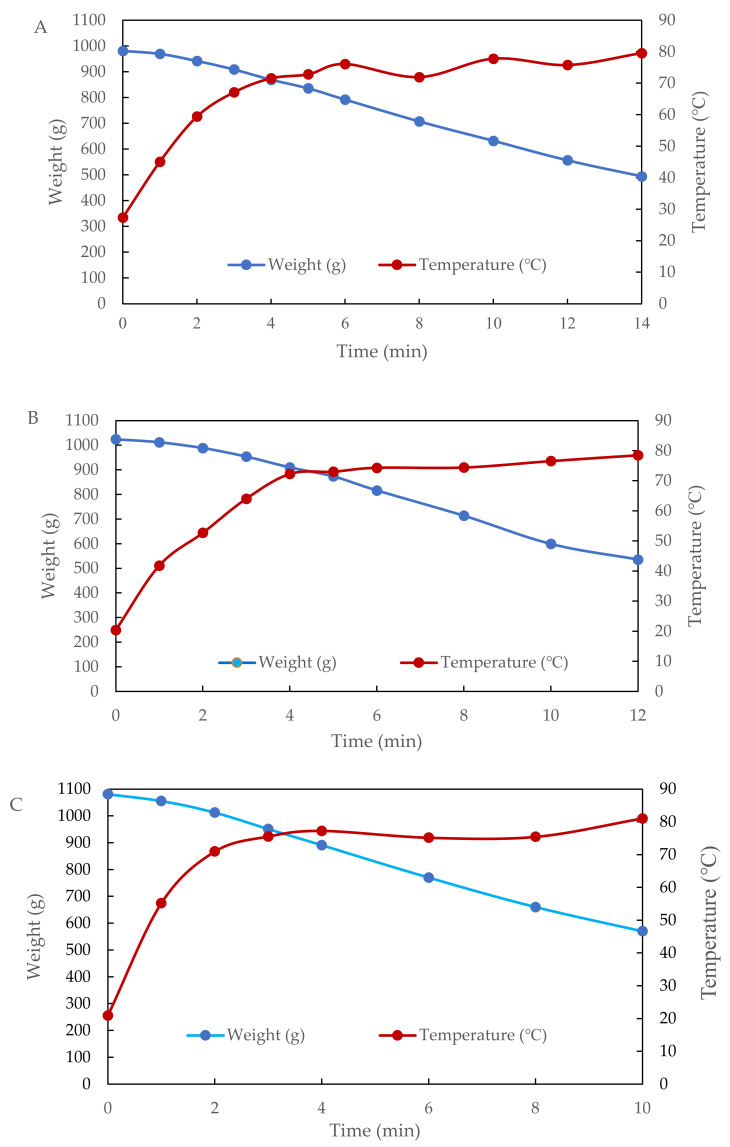
Temperature profiles and drying curves of wild bitter gourd extract mixed with soybean fiber: (**A**) 2:1, (**B**) 2:1.1, and (**C**) 2:1.2 during HARF heating.

**Table 1 foods-11-01173-t001:** Effect of focused ultrasonic extraction (FUE) and hot water extraction (HWE) on yield, active components, and antioxidant activity of wild bitter gourd.

Extraction Method	FUE	HWE
Extraction temperature (°C)	70	100
Extraction time (min)	10	60
Extraction yield (%)	22.4 ± 0.1 *	21.0 ± 0.0
Total saponins (mg/g)	0.288 ± 0.002	0.296 ± 0.022 *
Total proteins (mg/g)	1.198 ± 0.025	1.302 ± 0.032 *
Scavenging DPPH ability (%)	79.2 ± 0.1 *	59.6 ± 0.1

Data are expressed as mean ± S.D. (*n* = 3). Means with * in the same row were significantly different (*p* < 0.05). Scavenging DPPH ability (%) of 10 mg/mL Vitamin C and BHA were 95.5 ± 0.1 and 95.3 ± 0.1, respectively.

**Table 2 foods-11-01173-t002:** Drying rate of wild bitter gourd extract (E) mixed with soybean fiber powder (F) by 10 kW radio frequency heating.

E:F	Loading (kg)	Linear Regression Equation	R^2^	Rate (g/min)
2:1	1.0	W = −36.881t + 1007.8	0.996	36.881
2:1.1	1.0	W = −43.836t + 1066.4	0.984	43.836
2:1.2	1.0	W = −53.968t + 1103.2	0.995	53.968

W is weight of sample (g), and t is drying time (min) in linear regression equation.

**Table 3 foods-11-01173-t003:** Total counts and mold and yeast counts for wild bitter gourd extract, soybean fiber powder, and RF product.

Wild Bitter Gourd Sample	Extract from Wild Bitter Gourd	Soybean Fiber Powder	RF-Dried Extract Fiber Product
	Total bacterial colonies		
CFU/mL	523,000	9400	30
Picture (1:10)	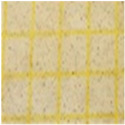	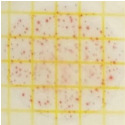	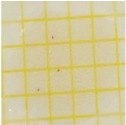
	Mold and yeast counts		
CFU/mL	>1000	30	<10
Picture (1:10)	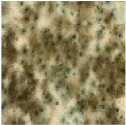	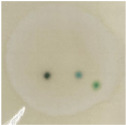	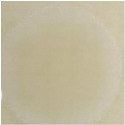

**Table 4 foods-11-01173-t004:** The changes of body weight (g) in mice for feeding RF-dried bitter gourd extract.

Group	0 Week	1 Week	2 Weeks	3 Weeks	4 Weeks
Normal	23.05 ± 1.64 ^b^	23.69 ± 1.65 ^a^	24.03 ± 1.63 ^a^	24.61 ± 1.21 ^a^	24.95 ± 1.57 ^a^
STZ	22.58 ± 0.88 ^ab^	22.94 ± 1.27 ^a^	23.67 ± 1.38 ^a^	24.01 ± 1.45 ^a^	24.78 ± 1.32 ^a^
STZ + Extract	21.69 ± 0.93 ^a^	23.66 ± 2.09 ^a^	24.16 ± 2.49 ^a^	24.90 ± 2.30 ^a^	25.35 ± 2.06^a^

Data are expressed as mean ± S.D. (*n* = 8). 0 week: after 2 weeks injecting nicotinamide and STZ; 1~4 weeks: free feeding for 1~4 weeks. ^a,b^ Means at different times with different superscript letter in the same column were significantly different (*p* < 0.05).

**Table 5 foods-11-01173-t005:** The changes of fasting blood glucose (mg/dL) in mice for feeding RF-dried bitter gourd extract.

Group	0 Week	4 Weeks
Normal	74.88 ± 6.20 ^b^	90.00 ± 6.50 ^c^
STZ	107.00 ± 6.21 ^a^	115.13 ± 5.69 ^a^
STZ + Extract	105.63 ± 4.34 ^a^	105.38 ± 4.07 ^b^

Data are expressed as mean ± S.D. (*n* = 8). 0 week: after 2 weeks of injection of nicotinamide and STZ; 4 weeks: free feeding for 4 weeks. ^a–c^ Means at different times with different superscript letter in the same column were significantly different (*p* < 0.05).

## Data Availability

The data and samples presented in this study are available on request from the corresponding author. Data is contained within the article.

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
