# Peer review of "Hot Air-Assisted Radio Frequency (HARF) Drying on Wild Bitter Gourd Extract"

_foods, 2022, doi:10.3390/foods11081173_

Round 1
Reviewer 1 Report
This is quite a limited work, in that only HARF has been used, but statements are made about other types of drying that cannot be supported in the context of the results of these experiments. This is a significant scientific error.
There is no significant evidence in the paper to support the statement (page 1) that:
“Therefore, the dried powder of ultrasonic wild bitter gourd extracts by hot air-assisted RF rather than freeze drying or spray drying had a hypoglycemic effect.”
and page 9:
“This suggests that the dried powder of ultra-sonic extracts by hot air-assisted RF rather than freeze drying or spray drying, had a hypoglycemic effect.”
There are some references to the literature to suggest that freeze drying gives higher quality products than spray drying. Even here (spray drying vs freeze drying), not all the literature shows that freeze drying is better than spray drying, so it is necessary for the authors to do a more thorough literature review.
However, the authors have not compared spray drying or freeze drying with HARF for their extracts in their work. Given the amount of biological variation within and between different extracts, unless the authors have directly compared their own extracts as spray dried or freeze dried by them with those that have been treated by HARF, then this statement is not acceptably supported and must be withdrawn.
Once this statement is withdrawn, the scope and comparative value of the paper is very unclear.
Author Response
Thank your (reviewer 1) comments.
We will respond to each suggestion individually, and the revised manuscript is attached.
- In abstract, “Therefore, the dried powder of ultrasonic wild bitter gourd
extracts by hot air-assisted RF rather than freeze drying or spray drying had a hypoglycemic effect.”
Answer: The sentence is changed to “Therefore, the dried powder of wild bitter gourd extracts by hot air-assisted RF had a hypoglycemic effect.”
- Page 9 “This suggests that the dried powder of ultra-sonic extracts by hot air-assisted RF rather than freeze drying or spray drying, had a hypoglycemic effect.”
Answer: The sentence is changed to “This suggests that the dried powder of ultrasonic extracts by HARF had a hypoglycemic effect.”
- There are some references to the literature to suggest that freeze drying gives higher quality products than spray drying. Even here (spray drying vs freeze drying), not all the literature shows that freeze drying is better than spray drying, so it is necessary for the authors to do a more thorough literature review.
Answer: The references 10-12 were related to spray drying of aqueous extracts and reference 13 was related to freeze-drying. These two drying methods are common in industry; however, they are expensive method, time and energy consuming. We want to provide a fast drying method for aqueous extracts by HARF.
- However, the authors have not compared spray drying or freeze drying with HARF for their extracts in their work. Given the amount of biological variation within and between different extracts, unless the authors have directly compared their own extracts as spray dried or freeze dried by them with those that have been treated by HARF, then this statement is not acceptably supported and must be withdrawn.
Answer: We withdraw the spray dried and freeze-dried sample statement. In conclusion, “As a result, the dried powder of ultrasonic wild bitter gourd extracts by HARF had a hypoglycemic effect.”
- In addition, in page 8, we add “In this study, the final temperatures of samples were also near 80°C, and the drying time was much less than that of spray drying and freeze-drying, because the feed rate of spray drying was only 10 mL/min [10] and freeze-drying required at least 24 hr.”
- We follow the editor’s comments and give the manuscript for English improvement.

Reviewer 2 Report
- Please check the citation and referencing as it is not consistent throughout the manuscript. For example, RF has been increasingly studied in food drying [14] (Zhou and Wang, 2019).
- Spray drying is an established drying for converting liquid to powder. The manuscript has not discussed the scope of spray drying. The performance of the proposed drying can be even compared with the yield from spray drying.
- Findings are not critically interpreted and compared with existing literature.
- Drying kinetics should be presented with Moisture content ( dry or wet basis).
- The quality of powder from pure extraction also needs to be analysed in order to observe the sole contribution of bitter melon in nutritional effect. Mixed with soybean fibre powder with different ratios definitely shows the effect of the concentration.
Author Response
Thank your (reviewer 2) comments.
We will respond to each suggestion individually, and the revised manuscript is attached.
(1) Please check the citation and referencing as it is not consistent throughout the manuscript. Answer: We check it and change “RF has been increasingly studied in food drying [14].”
(2) Spray drying is an established drying for converting liquid to powder. The manuscript has not discussed the scope of spray drying. The performance of the proposed drying can be even compared with the yield from spray drying.
Answer: Spray drying is common in industry; however, it is expensive method and time consuming. In page 8, we add “In this study, the final temperatures of samples were also near 80°C, and the drying time was much less than that of spray drying and freeze-drying, because the feed rate of spray drying was only 10 mL/min [10] and freeze-drying required at least 24 hr.”
(3) Findings are not critically interpreted and compared with existing literature.
Answer: It is difficult to compare with the literature because most current research involves the water extract by spray drying or freeze drying, and no use of HARF drying water extraction has been discovered.
(4) Drying kinetics should be presented with Moisture content (dry or wet basis).
Answer: The drying kinetics in Table 2 was shown weight loss during drying time. They appeared the constant drying rate period which were the zero order reaction. We add “ W is weight of sample (g), and t is drying time (min) in linear regression equation.” in the bottom of the table “
(5) The quality of powder from pure extraction also needs to be analyzed in order to observe the sole contribution of bitter melon in nutritional effect. Mixed with soybean fiber powder with different ratios definitely shows the effect of the concentration.
Answer: We only analyzed total saponins, total proteins and scavenging DPPH ability of the extract form bitter gourd by focused ultrasonic extraction (FUE) and hot water extract (HWE) in Table 1. Yes, mixed with soybean fiber powder with different ratios definitely shows the effect of the concentration. We used1 kg of concentrated bitter gourd extract was mixed with soybean fiber powder at a ratio of 2:1.1, which was dried by HARF 12 min for preparing feed with 5% RF dried bitter gourd to check hypoglycemic effect.
- We follow the editor’s comments and give the manuscript for English improvement.

Round 2
Reviewer 1 Report
The authors have withdrawn the unacceptable statement about other types of drying, given that there is no evidence to support their previous statement:
“This is quite a limited work, in that only HARF has been used, but statements are made about other types of drying that cannot be supported in the context of the results of these experiments. This is a significant scientific error.
There is no significant evidence in the paper to support the statement (page 1) that:
“Therefore, the dried powder of ultrasonic wild bitter gourd extracts by hot air-assisted RF rather than freeze drying or spray drying had a hypoglycemic effect.”
and page 9:
“This suggests that the dried powder of ultra-sonic extracts by hot air-assisted RF rather than freeze drying or spray drying, had a hypoglycemic effect.”
There are some references to the literature to suggest that freeze drying gives higher quality products than spray drying. Even here (spray drying vs freeze drying), not all the literature shows that freeze drying is better than spray drying, so it is necessary for the authors to do a more thorough literature review.
However, the authors have not compared spray drying or freeze drying with HARF for their extracts in their work. Given the amount of biological variation within and between different extracts, unless the authors have directly compared their own extracts as spray dried or freeze dried by them with those that have been treated by HARF, then this statement is not acceptably supported and must be withdrawn.
Once this statement is withdrawn, the scope and comparative value of the paper is very unclear.”
However, the scope and comparative value of the paper are still unclear. The authors need to either generate some evidence, or find some clear evidence in the literature, to compare their work with other drying methods.
Author Response
Thank your comment and suggestion. We answer in the attach file.
Freeze-drying or spray-drying are common methods for drying extracts in the health food industry or traditional Chinese medicine industry. However, this article provides another alternative drying method, which is to use the time-saving and energy-saving HARF to simultaneously dry and pasteurize the extract, and its hypoglycemic effect is also proved by STZ mouse experiments.
